# Enhancing Chemical Stability through Structural Modification of Antimicrobial Peptides with Non-Proteinogenic Amino Acids

**DOI:** 10.3390/antibiotics12081326

**Published:** 2023-08-17

**Authors:** Takahito Ito, Natsumi Matsunaga, Megumi Kurashima, Yosuke Demizu, Takashi Misawa

**Affiliations:** 1National Institute of Health Sciences, 3-25-26, Tonomachi, Kawasaki-shi 210-9501, Japan; w225501d@yokohama-cu.ac.jp (T.I.);; 2Graduate School of Medical Life Science, Yokohama City University, 1-7-29, Yokohama 230-0045, Japan

**Keywords:** cationic α,α-disubstituted amino acids, antimicrobial peptides, secondary structure, digestion tolerance

## Abstract

Multidrug-resistant bacteria (MDRB) remain a significant threat to humanity, resulting in over 1.2 million deaths per year. To combat this problem effectively, the development of therapeutic agents with diverse mechanisms of action is crucial. Antimicrobial peptides (AMPs) have emerged as promising next-generation therapeutics to combat infectious diseases, particularly MDRB. By targeting microbial membranes and inducing lysis, AMPs can effectively inhibit microbial growth, making them less susceptible to the development of resistance. Numerous structural advancements have been made to optimize the efficacy of AMPs. Previously, we developed 17KKV-Aib, a derivative of the Magainin 2 (Mag2) peptide, by incorporating a,a-disubstituted amino acids (dAAs) to modulate its secondary structure. 17KKV-Aib demonstrated potent antimicrobial activity against Gram-positive and Gram-negative bacteria, including multidrug-resistant *Pseudomonas aeruginosa* (MDRP), with minimal hemolytic activity against human red blood cells. However, 17KKV-Aib faces challenges regarding its susceptibility to digestive enzymes, hindering its potential as an antimicrobial agent. In this study, we designed and synthesized derivatives of 17KKV-Aib, replacing Lys residues with 4-aminopiperidine-4-carboxylic acid (Api), which is a cyclized dAA residue possessing cationic properties on its side chain. We investigated the impact of Api substitution on the secondary structure, antimicrobial activity, hemolytic activity, and resistance to digestive enzymes. Our findings revealed that introducing Api residues preserved the helical structure and antimicrobial activity and enhanced resistance to digestive enzymes, with a slight increase in hemolytic activity.

## 1. Introduction

Infectious diseases are considered a threat to humanity, and the development of antibiotics, such as penicillin, has saved many lives. However, with the inappropriate and prolonged use of antibiotics, the generation of multidrug-resistant bacteria (MDRB) that exhibit resistance to the drugs has become a problem. In recent years, more than 1.2 million people have died annually from MDRB such as multidrug-resistant *Pseudomonas aeruginosa* (MDRP) and methicillin- and vancomycin-resistant *Staphylococcus aureus* (MRSA and VRSA), which is still a major threat to humanity [1]. Therefore, the development of therapeutic agents based on different mechanisms of action is urgently required. Recently, antimicrobial peptides (AMPs) have received considerable attention as the next-generation drugs for infectious diseases [2,3]. To date, several AMPs have been reported, and their mechanism of action has been investigated [4,5,6,7,8]. Studies on their mechanism of action demonstrated that amphipathicity composed of hydrophobic and cationic amino acid residues and the formation of stabilized secondary structures are required to exert their antimicrobial activity by insertion into the microbial membrane, resulting in lysis of the bacteria [9]. Previously, we developed antimicrobial peptides based on Magainin 2 (Mag2), which is a representative AMP with a helical structure, sequenced by modulating their secondary structures [10,11,12]. Among them, we identified 17 N-terminal amino acid residues of the Mag2 sequence that are essential for antimicrobial activity. We enhanced the helical structure formation and antimicrobial activity by introducing 2-aminobutyric acid (Aib) residues, a representative α,α-disubstituted amino acid (dAA) that acts as a helical structure inducer. As a result, the peptide 17KKV-Aib (**1**) showed the potent antimicrobial activity against Gram-positive and Gram-negative bacteria, even MDRP, without significant hemolytic activity [10]. Despite exhibiting excellent antimicrobial activity, peptide **1** faces challenges due to its susceptibility to degradation by digestive enzymes, hindering its potential application as AMP for therapeutic use against MDRB. To address this issue, we investigated two strategies for enhancing the structural integrity of peptide **1**: (i) further restriction of the helical structure through the introduction of cyclized dAA residues and side-chain stapling, and (ii) replacement of lysine residues with other cationic amino acids. Cyclized dAA residues such as 1-aminocyclopentanoic acid (Ac_5_c) and the hydrocarbon side-chain stapling between two (*S*)-pentenylalanine (S_5_) residues at the *i* and *i +* 4 position exhibited stronger effects on helical structure stabilization, leading to increased antimicrobial activity when introduced into the peptide **1** sequence [11]. On the other hand, we observed that replacing lysine residues with cationic amino acids such as ornithine (Orn), diaminobutanoic acid (Dab), and arginine (Arg) resulted in comparable antimicrobial activity against Gram-positive and Gram-negative bacteria, but higher hemolytic activity occurred when the Arg residues were introduced instead of Lys residues. It is well known that the guanidino group of Arg residue shows higher basicity and affinity to the human cell membrane compared to amine group of Lys residue; hence, the hemolytic activity was increased. Moreover, peptides containing Orn and Dab residues, being non-proteinogenic amino acids, demonstrated tolerance against digestive enzymes [12]. These data suggested that the Lys residues in peptide **1** are replaceable with alternative cationic amino acid residues. Based on these findings, we hypothesized that substituting lysine residues with cyclized dAA possessing cationic moieties would enhance helical structure stability and improve chemical stability against digestive enzymes. Previous studies have shown that introducing 4-aminopiperidine-4-carboxylic acid (Api) residues, which are piperidine-type dAA residues, enhances the helical structure stability and water solubility [13,14,15]. Therefore, we designed and synthesized derivatives of AMPs based on 17KKV-Aib (**1**) by introducing Api residues (Figure 1). We investigated the effects of these Api residues on antimicrobial activity, hemolytic activity against human red blood cells (RBCs), and resistance to digestive enzymes.

## 2. Results

### 2.1. Synthesis of Fmoc-Protected Api 

Firstly, the Fmoc-protected Api was synthesized using a previously reported method (Figure 1) [16,17]. Briefly, starting from 4-piperidone monohydrate hydrochloride, we obtained α-Fmoc-protected Api via Bucherer–Bergs reaction, Boc-protection, hydrolysis of hydantoin, and Fmoc-protection with Fmoc-Cl. The NMR chart of Fmoc-Api(Boc)-OH is shown in Appendix A.

### 2.2. Design and Synthesis of AMPs Containing Api Residue

The chemical stability of 17KKV-Aib (**1**) was investigated using a digestion assay with proteinase K, which is a widely used representative protease for assessing sensitivity to digestive enzymes [18]. Proteinase K is widely recognized for its ability to cleave peptide bonds on the carboxylic sides of aliphatic, aromatic, or highly hydrophobic amino acids. Peptide **1** was treated with proteinase K for 24 h, and the remaining peptide fragments were evaluated using liquid phase chromatography–mass spectrometry (LC-MS) analysis. The results revealed that the cleavage occurred between the fifth Phe and sixth Leu residues of peptide **1** upon proteinase K treatment (Figure 2). These findings suggested that incorporating non-proteinogenic amino acids near the cleavable site could be a promising strategy for improving the chemical stability of peptide **1**. Based on these results, we designed and synthesized a series of derivatives, **2**–**8**, of 17KKV-Aib (**1**) by replacing each Lys residue with an Api residue to enhance the chemical stability of the peptides. The sequences of the synthesized peptides are listed in Table 1. All the peptides were synthesized using Fmoc-assisted solid-phase methods by Liberty Blue. The peptides were cleaved from the resin using a trifluoroacetic acid (TFA) cocktail, purified by reversed-phase high-performance liquid chromatography (HPLC) and identified by LC-MS. 

### 2.3. Preferred Secondary Structural Analysis by CD Spectra

Secondary structural analysis of the synthesized peptides **1**–**8** was performed using circular dichroism (CD) spectra in a 20 mM phosphate buffer solution (pH 7.4) with 1% sodium dodecyl sulfate (SDS), which has been reported to mimic the environment close to the cell membrane [19]. As shown in Figure 3, peptides **1**–**8** exhibited negative maxima at approximately 208 and 222 nm [20], indicating the formation of a stable α-helical structure. These results indicated that introducing the Api residue into the peptide **1** sequence did not significantly affect the secondary structures of peptides **2**–**8**. Because the presence of two Aib residues already stabilized the helical structure of peptide **1**, we can infer that the contribution of Api to further stabilization of the helical structure was relatively small.

### 2.4. Antimicrobial Activity of the Synthesized Peptides

The antimicrobial activities of peptides **1**–**8** against Gram-positive *Staphylococcus aureus*, Gram-negative *Escherichia coli* DH5α, *Pseudomonas aeruginosa*, and multidrug-resistant *P. aeruginosa* (MDRP) were evaluated. The minimum inhibitory concentration (MIC) against these bacteria was measured as an indicator of antimicrobial activity [21], and the results are discussed in Table 2. As previously reported, peptide **1** exhibited antimicrobial activity against Gram-positive and Gram-negative bacteria [8]. In contrast, the 17KKV-Aib derivatives **2**–**8**, which contained Api residues instead of Lys residues in the 17KKV-Aib sequence, demonstrated comparable levels of antimicrobial activity. This suggests that the introduction of Api residues only slightly impacted the antimicrobial activity of 17KKV-Aib derivatives. Peptides **2** and **3**, which incorporated an Api residue at the third and fourth positions, respectively, exhibited a twofold increase in antimicrobial activity against MDRP (MIC values of 6.25, 3.13, and 3.13 μM for peptides **1**, **2**, and **3**, respectively). However, peptides **2** and **3** showed half the activity against *E. coli* compared to that of peptide **1**.

### 2.5. Hemolysis Activity of the Synthesized Peptides against Red Blood Cells

AMPs primarily target microbial membranes, leading to their antimicrobial activity through membrane lysis. However, it is important to consider their potential effects on human cell membranes, as this could raise concerns regarding cytotoxicity [23]. To evaluate the cytotoxicity, we investigated the hemolytic activity of the tested peptides, **1**–**8**, using human red blood cells. Hemolytic activity was assessed by measuring the absorption at 535 nm, which indicates the leakage of red blood cell cytoplasmic contents and hemoglobin. The results, shown in Table 2 and Appendix A, indicate that peptides **2**–**4** and **7** exhibited hemolytic activity at 12.5 and 25 μM, respectively. In contrast, peptides **5**, **6**, and **8** demonstrated similar levels of hemolytic activity to peptide **1**. These findings suggest that introducing the Api residue tends to increase the hemolytic activity of the peptides.

### 2.6. Chemical Stability of the Synthesized Peptides against Digestive Enzymes

We conducted a tolerance assessment using digestive enzymes to confirm the chemical stability of peptides **1**–**8**. As previously mentioned, the peptides were subjected to a digestion assay using proteinase K, and the resulting peptide fragments were analyzed using LC-MS. As shown in Figure 4, peptides **2**–**4**, which contained Api residues near the cleavable site, exhibited a longer half-life than peptide **1**. After 24 h of treatment, the residual amount exceeded 60%. In contrast, peptides **5**–**8**, which had Api residues near Aib residues, displayed similar chemical stability against proteinase K as peptide **1**. The predicted cleavage site and the MS spectra of fragments of each peptide **1**–**8** are shown in Appendix A. The fragments analysis of peptides **2**–**4**, which have Api residue around cleavable site, revealed that the site between 16th Phe and 17th Lys residues could also be cleaved by proteinase K after 24 h incubation. We also investigated the chemical stability of peptide **2** with or without fetal bovine serum (FBS), as shown in Appendix A. Peptide **2** was stable in PBS and 70% of peptide **2** remained after 24 h incubation with 50% FBS in PBS, indicating that peptide **2** showed tolerance against digestive enzymes in the presence of serum. These findings indicate that introducing the Api residue around the cleavable site prolonged the half-life of the peptides. However, the successive introduction of non-proteinogenic amino acids did not significantly affect the chemical stability of the peptides.

### 2.7. Calculation Analysis of Charge Surface with or without Api Residue

Finally, we performed the calculation analysis of tested peptides **1**, **2**, and **8** to investigate the effects of Api residue on the charge surface. As shown in Figure 5, the cationic surface area, as shown in blue, were decreased with the introduction of Api instead of Lys residue, indicating that the substation of Lys with Api decreased the cationic surface of peptides and increased the relative hydrophobicity. Therefore, the tested peptides containing Api residues tend to increase the hemolytic activity, expect for peptide **8**. Peptide **8** has Api residue on the C-terminal, and the calculation analysis demonstrated that the anionic surface area, colored red, of peptide **8** was relatively increased by substitution of Lys residue with Api compared to other derivatives. Therefore, we hypothesized that the affinity of peptide **8** to the cell membrane was decreased; hence, a significant increase in hemolytic activity did not occur. Calculation analysis revealed that the substitution of Lys residues with Api residues increased the amphipathicity of whole peptides, resulting in increased hemolytic activity against RBCs.

## 3. Conclusions

In this study, we designed and synthesized AMP derivatives **2**–**8** based on 17KKV-Aib (**1**) by incorporating Api, a cationic non-proteinogenic amino acid. We evaluated the effects of Api on various biological activities, including the secondary structure, antimicrobial activity, hemolytic activity, and chemical stability. Peptides **2**–**8** displayed comparable CD spectra intensity to peptide **1**, suggesting a formation of the a-helical structure similar to peptide **1**. Incorporation of Api residues into the peptide sequence did not further stabilize the helical structure. Furthermore, peptides **2**–**8** exhibited similar antimicrobial activity against Gram-positive and Gram-negative bacteria to that of peptide **1**. Notably, peptides **2** and **3**, which included an Api residue at the third and fourth positions, respectively, displayed a two-fold increase in antimicrobial activity against MDRP (MIC values of 6.25, 3.13, and 3.13 μM for peptides **1**, **2**, and **3** respectively). These results suggested that the introduction of the Api residue had minimal effects on the secondary structure and antimicrobial activity. However, it led to a slight increase in hemolytic activity, up to fourfold. The exact reason behind the higher hemolytic activity observed in peptides containing Api residues remains unclear. However, it is hypothesized that the disparity side chain basicity between Lys and Api residues may impact the hemolytic activity, similar to how substitution with Arginine enhances hemolytic activity. Moreover, the digestion assay revealed that the sites between 5th Phe and 6th Leu and 16th Phe and 17th Lys residues of peptide **1** were sensitive for the cleavage by treatment of proteinase K. On the other hand, the presence of Api residues near the cleavable site interrupts peptide degradation due to digestive enzymes. Peptides **2**–**3**, in particular, are anticipated to exhibit higher and longer-lasting antimicrobial activity against MDRP than peptide **1**. Api is an alternative to lysine residues and can potentially develop helical peptides with cationic side chains. Our findings suggest that incorporating specific non-proteinogenic amino acids at cleavable sites targeted by digestive enzymes can contribute to developing bioactive peptides with enhanced chemical stability. Moreover, the calculation analyses were conducted to investigate the effects of the introduction of Api residue on their surface charge area. As shown in Figure 5, the introduction of Api residue instead of Lys residue decreased the cationic surface area of peptides, resulting in enhancement of amphipathicity and hemolytic activity. Therefore, further structural development that does not reduce the cationic surface area would be expected to provide similar chemical stability without increasing the hemolytic activity. The introduction of Api derivatives containing alkylamine moiety on the nitrogen atom would be a promising strategy to develop novel AMP derivatives.

## 4. Materials and Methods

### 4.1. General Information

All chemicals were purchased from Watanabe Chemical Industries, LTD. (Hiroshima, Japan), Sigma-Aldrich Co., LLC (St. Louis, MO, USA), Kanto Chemicals Co., Inc. (Tokyo, Japan), Tokyo Chemical Industry Co., Ltd. (Tokyo, Japan), and FUJIFILM Wako Pure Chemical Industries Ltd. (Osaka, Japan), and they were used without further purification. Mass spectra were obtained using a Shimadzu IT-TOF MS instrument equipped with an electrospray ionization source. ^1^H NMR spectra were measured on an ECZ 600R spectrometer (JEOL) using deuterated solvents. Chemical shift values (ppm) are expressed in δ (ppm) with the residual solvent peak (7.26 for CDCl_3_) as the internal standard.

### 4.2. Synthesis of Api

The Fmoc protected Api was synthesized according to a previous report [17]. Boc-protected hydantoin **A** (10.0 g, 21.3 mmol) derived from 4-piperidone monohydrate was dissolved in THF (70 mL), and 2 M NaOH aq (70 mL) was added to the solution. After stirring for 6 h at room temperature, the organic layer and aqueous layer were separated. The aqueous layer was cooled to 0 °C and 6 M HCl aq. was added until the pH reached to 6~7 to obtain the precipitate. The precipitate was collected by filtration and dried under reduced pressure overnight at 60 °C. Then, the precipitate was suspended in CH_2_Cl_2_ and stirred for 45 min. The solution was filtered and the remaining white solid was dried under reduced pressure at 80 °C overnight to yield **B** (2.71 g, 52%).

### 4.3. Synthesis of Fmoc-Protected Api

Compound **B** (5.42 g, 22.2 mmol) was dissolved in CH_2_Cl_2_ (160 mL) and *N*,*N*-diisopropylethylamine (9.67 mL, 55.5 mmol) was added to the solution. After stirring for 30 min at room temperature, trimethylsilyl chloride (5.67 mL, 44.4 mmol) was added to the reaction mixture and refluxed. After 1 h, an extra portion of trimethylsilyl chloride (5.67 mL, 44.4 mmol) was added to the reaction mixture. N_2_ gas was blown into the flask to remove HCl gas several times during the reaction. After 1 h, the reaction mixture was cooled to 0 °C and Fmoc-Cl (5.74 g, 22.2 mmol) was added to the reaction mixture. The mixture was stirred for 2 h at 0 °C, then concentrated under reduced pressure. The resulting sticky paste was dissolved in ether and washed with sat. NaHCO_3_ aq. 1M HCl aq was added to the aqueous layer until the pH reached 2 and was extracted with CH_2_Cl_2_ three times. The combined organic layer was dried over Na_2_SO_4_ and concentrated under reduced pressure. The resulting paste was purified by silica gel column chromatography (EtOAc:*n*-hexane = 1:4 to 2:3 containing 1% acetic acid). The purified product was dissolved in toluene and concentrated under reduced pressure to remove the remaining acetic acid to obtain Fmoc-Api(Boc)-OH (10.8 g, quant) as a white solid.

^1^H NMR (600 MHz, CDCl_3_) δ 7.75 (d, *J* = 7.2 Hz, 2H), 7.57 (d, *J* = 7.2 Hz, 2H), 7.40 (t, *J* = 7.2 Hz, 2H), 7.31 (t, *J* = 7.2 Hz, 2H), 5.03 (br, 1H), 4.47 (br, 2H), 4.21 (s, 1H), 3.81 (br, 2H), 3.07 (br, 2H), 2.08 (br, 4H), 1.46 (s, 9H). HRMS (ESI): *m*/*z* calcd. for C_21_H_23_N_2_O_4_ [M + H^−^ Boc]^+^ 367.1652, found 367.1644.

### 4.4. Peptide Synthesis

The designed peptides were synthesized using Fmoc-based solid-phase methods on Rink Amide Pro Tide Resin (LL) (CEM) via Liberty Blue Automated Microwave Peptide Synthesizer (CEM). A representative coupling and deprotection cycle are described as follows: 250 mg Rink Amide Pro Tide Resin (LL) (loading: 0.2 mmol/g) was soaked for 30 min in dichloromethane. After the resin was washed with *N*,*N*-dimethylformamide (DMF), amino acids (5 eq.), Oxyma Pure (10 eq.) and DIC (10 eq.) dissolved in a solution of DMF were added to the resin. Fmoc protective groups were deprotected using 20% piperidine (PPD) in DMF. The resin was suspended in a cleavage cocktail (95% TFA, 2.5% water, 2.5% triisopropylsilane (TIPS)) at room temperature for 3 h. The TFA was removed to a small volume with a stream of N_2_ and dripped into cold ether to precipitate the peptide. The precipitates were dissolved in dimethylsulfoxide (DMSO) and purified by reverse-phase HPLC using a Discovery BIO Wide Pore C18 column (5 µm, 25 cm × 21.2 mm, solvent A: 0.1% TFA/water, solvent B: 0.1% TFA/MeCN, flow rate: 10.0 mL/min, gradient: 10–90% gradient of solvent B over 40 min). After purification, the solution was lyophilized to obtain target peptides. The peptide purity was assessed by analytical HPLC using InertSustainSwift C18 column (3 µm, 25 cm × 4.6 mm; solvent A: water containing 0.1% TFA, solvent B: MeCN containing 0.1% TFA, flow rate: 1.0 mL/min, gradient: 10–90% gradient of solvent B over 30 min). HPLC charts and mass spectrometric data for each peptide are shown in the Appendix A.

### 4.5. CD Spectrometry

The synthesized peptides **1**–**8** were dissolved in 20 mM phosphate buffer (pH 7.4) containing 1% SDS at a concentration of 100 µM. The CD spectra for each peptide were measured using JASCO J-720W with the following parameters: wavelength: 190–260 nm, bandwidth: 1 nm, response: 1 s, speed: 20 nm min^−1^, accumulations: 3 times. The data are expressed in terms of [θ], that is, the total molar ellipticity (deg cm^2^ dmol^−1^).

### 4.6. Antimicrobial Activity

One type of Gram-positive bacteria, *S. aureus* NBRC 13276, and three Gram-negative bacteria, *E. coli* DH5α, *P. aeruginosa* NBRC 13275, and MDRP, were selected for the evaluation of the antimicrobial activity of the synthesized peptides. The selected bacterial strains were obtained from the Biological Resource Center, NITE (NBRC; Tokyo, Japan), and *Escherichia coli* DH5α cells were purchased from BioDynamics Laboratory, Inc. (Tokyo, Japan). The antimicrobial activities of the tested peptides **1**–**8** against each stain were examined by the standard broth microdilution method, as previously reported [22]. Briefly, the bacteria were seeded and grown overnight at 37 °C on Agar media for other organisms i (Agar medium) and then collected with the media for other organisms ii (liquid medium), according to the 17th revision of *Japanese Pharmacopoeia*. Next, 10 μL of the peptide solution per well was added to each well of a sterile 96-well plate, and 90 μL per well of inoculation was mixed with 10^4^ CFUs (colony forming units) per mL and added to each well, and the plate was incubated for 18 h at 35 °C. The MIC was defined as the lowest concentration of peptide that completely inhibited bacterial growth.

### 4.7. Hemolysis Activity

Human red blood cells (RBCs) were supplied by the Japanese Red Cross Society (Tokyo, Japan). The hemolytic assay of the synthesized peptides was performed as previously reported [24]. Briefly, 5.0 mL of the total erythrocytes were washed three times with 172 mM Tris-HCl buffer (pH = 7.6, wash buffer:WB) and 1.0 mL of precipitated erythrocytes was diluted with 9.0 mL of WB. Then, 0.5 mL of the suspension was diluted in WB, and 50 µL of RBC solution was incubated with 50 µL of each peptide for 30 min at 37 °C (final peptide conc. from 100 to 0.39 μM). The suspensions were then centrifuged at 400 rpm for 5 min. The absorbance of the supernatant was measured at a wavelength of 535 nm. M-Lycotoxin [22], which is a typical hemolytic toxic peptide derived from Wolf spider, was used as a positive control, and the absorbance of the sample from DMSO and M-Lycotoxin treatment was defined as 0 and 100%, respectively.

### 4.8. Digestion Assay

The resistance of the synthesized peptides to digestive enzymes was evaluated using Proteinase K, which exhibits a broad proteolytic spectrum. Each peptide (1 mM) was treated with 0.002% (*v*/*v*) Proteinase K, and the mixture was incubated at 37 °C for 2–24 h. The enzymatic reaction was quenched by the addition of an acidic solution (0.1% TFA/H_2_O/MeCN). The residual peptides were assessed by analytical HPLC using an InertSustainSwift C18 column (3 µm, 25 cm × 4.6 mm, solvent A: water containing 0.1% TFA, solvent B: MeCN containing 0.1% TFA, flow rate: 1.0 mL/min, gradient: 10–90% gradient of solvent B over 30 min). The digestion assay was performed in two independent experiments, and the average values from them were calculated.

### 4.9. Calculation Analysis of Charge Surface

Charge surface calculations were performed on MOE 2022.07 software (CCD, Canada) based on the minimized peptide structure. The optimization of peptide conformation was carried out via molecular mechanics (MM) with fixing of its structure as an a-helical structure. The force fields were set to AMBER-10.

## Data Availability

Data are available from the authors on reasonable request. The HPLC analytical method and characterization are outlined in the Appendix A.

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
