# Peer review of "Enhancing Chemical Stability through Structural Modification of Antimicrobial Peptides with Non-Proteinogenic Amino Acids"

_antibiotics, 2023, doi:10.3390/antibiotics12081326_

Round 1

Reviewer 1 Report

The inappropriate and prolonged use of antibiotics causes the generation of multidrug-resistant bacteria which exhibit resistance to the drugs has become a problem. To overcome this problem, antimicrobial peptides have been considered as the next-generation drugs for infectious diseases. The incorporation of new elements, such as non-nature amino acids and cyclized rings on the peptides, could usually increase the peptide stability and antibacterial activity. In this paper, the authors mutated lysine residues with the special amino acid Api at different positions of the antimicrobial peptides 17KKV-Aib, resulting in seven derivatives of this peptide. Their findings suggest that incorporating specific non-proteinogenic amino acid Api at cleavable sites prolongs the half-life of the peptides under proteinase K. However, the introduction of Api neither enhance the peptide stability nor improve its antibacterial activity. Besides, the peptides with Api residue tend to increase the hemolytic activity of the peptides. Taking it all together, the incorporation of Api didn’t show a significant advantage to improve and engineer this antibacterial peptide. Furthermore, the manuscript didn’t show creative work of antibacterial peptide modification. This manuscript doesn’t match the level to be published in the journal Antibiotics. I would reject this manuscript.   

Author Response

Thanks for your rigorous review of our manuscript.  Based on the reviewer's comments, we revised the manuscript and added the data. We would be happy to review the revised manuscript again. 

Reviewer 2 Report

The authors described enhanced stability of antimicrobial peptides through incorporation of Api at cleavable sites. The manuscript can be published with the following edits.

1) Add subdivisions under 'Results' to describe individual studies. For example, 2.1 Circular Dichroism etc.

2) The authors must add a future direction at the conclusion.

3) Provide schemes for synthesis section.

Author Response

Reviewer 2

  • Add subdivisions under 'Results' to describe individual studies. For example, 2.1 Circular Dichroism etc.

In accordance with the reviewer’s comment, the subdivisions were added as yellow highlighted.

  • The authors must add a future direction at the conclusion.

In the accordance with the reviewer’s comment, the future direction such as further structural development was described in conclusion section in page 8 line 244.

Moreover, the calculation analyses were conducted to investigate the effects of introduction of Api residue on their surface charge area. As shown in Figure 4, the introduction of Api residue instead of Lys residue decreased the cationic surface area of peptides, resulting in enhancement of amphipathicity and hemolytic activity. Therefore, a further structural development that does not reduce the cationic surface area would be expected to provide similar chemical stability without increase of hemolytic activity. The introduction of Api derivatives containing the alkylamine moiety on the nitrogen atom would be promising strategy to develop the novel AMP derivatives.(ChemBioChem, 2015, 17, 137.)

3) Provide schemes for synthesis section.

In accordance to the reviewer’s comment, the synthetic scheme of the Fmoc-protected Api was provided as scheme 1 in Page 3 line 91.

Reviewer 3 Report

The manuscript provides an extensive focus on exploring chemical stability of antimicrobial peptides through Structural Modification in associationship with Non-Proteinogenic Amino Acids. The work is reasonably written, and slightly discussed, and the data are not much conceiving with the aspects of stability of the peptides.

Thus, the article won’t be suitable to attract high attention from the readers of Antibiotics due to poor data representation and unavailability of very important stability experiments. As such, I recommend acceptance after considering the suggestions as described above and below:

Ø  I suggest authors to look carefully and describe the number of experiments performed and represent it with standard deviation in the figure 3.

Ø  Other stability experiments are compulsory essential in Normal Saline, Phosphate Buffer, and 50% FBS to understand the main consequences related to the stability of Antimicrobial Peptides with biological system.

Ø  Authors have highlighted about the 1H NMR studies, thus suggest including its spectra in the main manuscript.

Ø  I also suggest authors to identify and compare the surface charge of their Antimicrobial Peptides before and after the Structural Modification with Non-Proteinogenic Amino Acids

Ø  I also request authors to showcase their Haemolytic data in graphical representation rather than table. This will really help readers to understand the significance of work.

Moderate editing of English language required!

Author Response

Reviewer 3

  • I suggest authors to look carefully and describe the number of experiments performed and represent it with standard deviation in the figure 3.

Thanks for the kind suggestion. The digestion assays were performed in two experiment and the average values of them were shown. Therefore, we added the sentence about the assay conditions in caption of Figure 3 and experimental section.

  • Other stability experiments are compulsory essential in Normal Saline, Phosphate Buffer, and 50% FBS to understand the main consequences related to the stability of Antimicrobial Peptides with biological system.

Thanks for the useful suggestions. As reviewer’s comments, the other stability experiment were important to investigate the stability of the peptides in solution or in blood.

Previously, Gellman reported that the GLP-1 analogues containing non-proteinogenic amino acids were developed to enhance the chemical stability. Based on their report, the peptide, which showed the tolerance against digestive enzymes in vitro, also showed the long lasting effects in vivo experiment. (Gellman et al. A potent a/b-peptide analogue of GLP-1 with prolonged action in vivo. J. Am. Chem Soc. 2014, 136, 12848.) Moreover, it has been reported that the chemical stability of AMPs was investigated using trypsin.(Bang et al. Int. J. Mol. Sci. 2020, 21, 3602.) Therefore, we performed the digestion assay using proteinase K, and expected that the peptides 1-3 would stable in solution and blood.

In accordance with reviewer’s comments, the chemical stability of peptide 2 in PBS was investigated as below. The further investigation of chemical stability of peptides in 50% FBS will be performed in future work.

0 h                                       24 h

  • Authors have highlighted about the 1H NMR studies, thus suggest including its spectra in the main manuscript.

Fmoc-Api has been already reported as ref 17 and 18, and its NMR spectrum was not considered necessary to discuss in the main text. Therefore, the 1H NMR chart was added in supplementary data as Figure S1.

  • I also suggest authors to identify and compare the surface charge of their Antimicrobial Peptides before and after the Structural Modification with Non-Proteinogenic Amino Acids

As reviewer’s comments, we performed the calculation of the surface charge of peptides 1, 2, and 8. The calculation results were added in main manuscript and Figure 4 in Page 6 line 200, and the calculation methods were also described in experimental section in page 10 line 365.

2.7 Calculation analysis of charge surface with or without Api residue.

Finally, we performed the calculation analysis of the tested peptides 1, 2, and 8, to investigate the effects of Api residue on the charge surface. As shown in Figure 4, the cationic surface area as shown blue color, were decreased by introduction of Api instead of Lys residue, indicating that the substation of Lys with Api decreased the cationic surface of peptides and increased the relative hydrophobicity. Therefore, the tested peptides containing Api residues tend to increase the hemolytic activity expect for peptide 8. The peptide 8 has Api residue on the C-terminal and the calculation analysis demonstrated that the anionic surface area colored red of peptide 8 was relatively increased by substitution of Lys residue with Api compared to other derivatives. Therefore, we hypothesized that the affinity of peptide 8 to cell-membrane was decreased, hence the significant increase of hemolytic activity was not occurred. The calculation analysis revealed that the substitution of Lys residues with Api residues increased the amphipathicity of whole peptides, resulting in the increased the hemolytic activity against RBCs.

Figure 4. Calculation analysis of surface charge of 1, 2 and 8. The blue and red color represents the cationic and anionic surface.

  • I also request authors to showcase their Haemolytic data in graphical representation rather than table. This will really help readers to understand the significance of work.

In accordance with the reviewer’s comment, the representative results of hemolytic assay were shown in Figure S2 as bar graph.

Round 2

Reviewer 1 Report

The inappropriate and prolonged use of antibiotics causes the generation of multidrug-resistant bacteria which exhibit resistance to the drugs has become a problem. To overcome this problem, antimicrobial peptides have been considered as the next-generation drugs for infectious diseases. The incorporation of new elements, such as non-nature amino acids and cyclized rings on the peptides, could usually increase the peptide stability and antibacterial activity. In this paper, the authors mutated lysine residues with the special amino acid Api at different positions of the antimicrobial peptides 17KKV-Aib, resulting in seven derivatives of this peptide. Their findings suggest that incorporating specific non-proteinogenic amino acid Api at cleavable sites prolongs the half-life of the peptides under proteinase K. However, the peptides with Api residue tend to increase the hemolytic activity of the peptides. The authors explained this by using calculation analysis, which showed that the introduction of Api residue instead of Lys residue decreased the cationic surface area of peptides, resulting in the enhancement of amphipathicity and hemolytic activity. In contrast, the introduction of Api that does not reduce the cationic surface area would be expected to provide similar chemical stability without an increase in hemolytic activity. The manuscript gave a detailed study and example of the incorporation of the unnatural amino acid Api to develop antimicrobial peptides. After being revised, now this manuscript could be accepted and published in the journal Antibiotics.

Author Response

Thank you for your fairly review. The manuscript was improved by reviewer's comments.

Reviewer 3 Report

I suggested authors to compulsory perform stability experiments in 50% FBS to understand the main consequences related to the stability of Antimicrobial Peptides when it interact with biological system especially blood and other vital organs.

Thus again request to performit to improve the showcase the of importance of their peptide.

I have also suggested authors to identify the surface charge of their peptide whose values (mV) can be easily estimated by Zeta Sizer. 

Needs a minor corrections.

Author Response

Thank you for your fairly review. We performed the additional experiment as reviewer's comments. The results were uploaded as word file. I would like you to review our revised version of manuscript again. 

Round 3

Reviewer 3 Report

I appreciate authors efforts for doing experiments.

Unfortunately, Authors fails to corelate their data with their experimental results and hypothesis. 

Still the manuscript has serious flaws!

It can be improved.